# The Underlying Physicochemical Properties and Starch Structures of *indica* Rice Grains with Translucent Endosperms under Low-Moisture Conditions

**DOI:** 10.3390/foods11101378

**Published:** 2022-05-10

**Authors:** Fei Chen, Yan Lu, Lixu Pan, Xiaolei Fan, Qianfeng Li, Lichun Huang, Dongsheng Zhao, Changquan Zhang, Qiaoquan Liu

**Affiliations:** 1Key Laboratory of Crop Genomics and Molecular Breeding of Jiangsu Province, Key Laboratory of Plant Functional Genomics of the Ministry of Education, Yangzhou University, Yangzhou 225009, China; feich12345@outlook.com (F.C.); luyan@yzu.edu.cn (Y.L.); panlixu288@163.com (L.P.); xlfan@yzu.edu.cn (X.F.); qfli@yzu.edu.cn (Q.L.); lchuang@yzu.edu.cn (L.H.); dszhao@yzu.edu.cn (D.Z.); cqzhang@yzu.edu.cn (C.Z.); 2Jiangsu Co-Innovation Center for Modern Production Technology of Grain Crops, State Key Laboratory of Hybrid Rice, Jiangsu Key Laboratory of Crop Genetics and Physiology, Yangzhou University, Yangzhou 225009, China

**Keywords:** rice, grain appearance, amylose content, starch structure

## Abstract

Rice grain quality is a complex trait that includes processing, appearance, eating, cooking, and nutrition components. The amylose content (AC) in the rice endosperm affects the eating and cooking quality along with the appearance of milled rice. In this study, four *indica* rice varieties with different ACs were used to study the factors affecting endosperm transparency along with the physical and chemical characteristics and eating quality of translucent endosperm varieties. Endosperm transparency was positively correlated with water content and negatively correlated with the cumulative area of cavities within starch granules. The *indica* landrace 28Zhan had a translucent endosperm and exhibited good taste. Based on starch fine structure analysis, long-chain amylopectin and the B2 chain of amylopectin might be major contributors to the good taste and relatively slow digestion of this landrace.

## 1. Introduction

Rice (*Oryza sativa* L.) is one of the most important food crops in the world. The improvement of living standards has led to demands for rice with better cooking, flavor, and appearance characteristics [1,2,3]. The amylose content within rice grains is the key factor determining rice quality and particularly the eating quality and endosperm transparency, which are important characteristics that determine the rice’s value as a commodity [1]. While amylose content is an important factor, it is not the only determinant of rice eating quality; the amylopectin fine structure, particularly the short chains (A chains and short B chains), contributes greatly to rice eating quality [4]. Therefore, the rice taste and endosperm appearance can be fine turned by controlling amylose content and the amylopectin fine structure. 

The appearance of rice is usually described based on the chalkiness, farina, and waxiness of rice [5,6,7]. Rice with a chalky endosperm often has a loose starch filling, a large gap between starch grains, and an uneven water content. In terms of grain transparency, rice varieties with low amylose content generally become dull with the loss of water during storage and processing [8]. Low transparency is detrimental for the grain appearance, especially in some high-quality rice with low amylose content. While the causes of rice transparency have been studied in *japonica* rice under different genetic backgrounds, the formation mechanism of rice opacity remains unclear [9,10]. Recently, Zhang et al. generated a series of rice lines with medium to low amylose levels under the same *japonica* genetic background and found that under certain conditions, the amount of water loss was positively correlated with the transparency of the rice endosperm [11]. Other studies have shown that soft *japonica* rice (with low amylose content) generally exhibits a dull or milky grain appearance after the rice grains are stored under normal conditions [12,13]. Similarly, Liu et al. reported a dark endosperm phenotype of a Yunnan soft rice variety (with a low amylose content of approximately 10%) [14]. However, there are few studies on the formation of translucent endosperm in *indica* rice. Further study is needed to optimize the trade-off between grain appearance and eating quality. 

Rice eating quality is a composite index that includes the smell, color, shape, taste, viscosity, hardness, and other sensory indicators after cooking [4]. The eating quality of rice is affected by many factors including amylose content, gelatinization temperature (GT), gel consistency, pasting viscosity, and texture [15]. In addition to the influence of amylose, the structure of starch and the distribution of starch chain length play important roles in rice eating quality. In addition, rice storage protein content and properties contribute to the differences in eating quality between varieties, especially for rice varieties with similar amylose levels [16]. Therefore, it is necessary to evaluate the relationships between various indexes and rice eating quality and appearance. 

It is well known that the amylose content is mainly controlled by the *Wx* gene, and the allelic variation of the *Wx* locus (*Wx^a^*, *Wx^b^*, *Wx^mp^*, *wx*, etc.) contributes greatly to the different amylose levels within rice varieties [11]. In previous studies, we collected numerous rice accessions with different grain qualities and obtained 28zhan (28Z), an *indica* rice landrace with a translucent endosperm and relatively low amylose content. Interestingly, we found that 28Z rice carries the *Wx^b^* allele which controls medium amylose level (~15%). Thus, the 28Z rice is a novel low-amylose type that distinguishes it from other low-amylose rice carrying *Wx^mp^* and *Wx^op^* alleles [11]. In the present study, our aim is to investigate factors affecting endosperm transparency in *indica* rice. Thus, one *indica* glutinous rice variety (carrying the non-functional *wx*) and two normal *indica* rice varieties (carrying *Wx^b^*) with intermediate AC were selected as control varieties for grain quality and starch structure analyses. The grain transparency, physical and chemical properties, eating quality, starch particle morphology, and fine structure of starch under low moisture content were investigated in detail. The findings improve our understanding of the factors affecting rice grain appearance and rice eating quality in *indica* rice. 

## 2. Materials and Methods

### 2.1. Rice Varieties and Planting Conditions

Four *indica* rice varieties (*O. sativa* L. subsp. *indica*) were used in the experiment: one glutinous rice variety, Yangfunuo 4 (YFN), and three non-glutinous varieties, 28Z, Huanghuazhan (HHZ), and Yangdao 6 (also known as 9311). These varieties are typical *indica* varieties, and their heading date and plant height are similar. They were planted in the experimental field of Yangzhou University (Jiangsu, China) from May to October of 2021. The above varieties were planted in triplicate and arranged randomly in each plot under the same management conditions. Mature seeds were harvested from each line at the end of October. 

### 2.2. Rice Flour and Starch Preparation and Sorption Properties Determination

After being stored in a cool and ventilated place, some of the harvested mature rice was roughened with a huller, refined, ground into a powder, and sieved through a 100-mesh sieve. The sieved rice flour was then placed into an oven at 40 °C for 2 d and the dampness at room temperature was balanced for 2 d to ensure the same water content in the experiment. Finally, the rice was sealed in a self-sealing bag for use. For rice starch extraction, 10 g of white rice was used for starch preparation using the alkaline protease method described by Li et al. [17]. For grain transparency analysis, some of the harvested mature seeds were dried directly in the oven (Model FDL115, Binder Ltd., Tuttlingen, Germany) at 40 °C for 2, 4, 6, 8, 12, and 24 h, milled into white rice, and stored in self-sealing bags. The moisture contents of the above rice flours and white rice were measured using a moisture analyzer (Mettler Toledo MJ33, Greifensee, Switzerland) according to the method of Li et al. [17].

### 2.3. Physical and Chemical Quality Analyses

The apparent amylose content (AAC) of rice flour was determined using the iodine colorimetry method according to the work of Tan et al. with some modifications [18]. Briefly, 20.0 mg samples were weighed in a 2 mL centrifuge tube, 0.1 mL ethanol was added into each tube and shaken gently. Then, 1.8 mL of 1 N NaOH solution was added and the tubes were kept in a shaker (60 °C) at 200 rpm for 2 h. Afterwards, 100 μL of the suspension were transferred into a 10 mL centrifuge tube with 9 mL distilled water, and then 200 μL sodium acetate solution (pH 4.3) and 200 μL iodine solution (0.02%) were added into the tube, and the absorbance was measured at 620 nm. The AAC was calculated from a standard curve developed by using amylose standards.

The taste value of cooked milled rice was measured using a Japanese rice taste analyzer (STA1B, Sasaki, Hiroshima, Japan) as described by Li et al. [17]. Specifically, 30 g rice samples were weighed in an aluminum container, rinsed three times, and water was added at a 1:1.12 water ratio (*w*/*w*). Afterward, the rice samples were steamed for 45 min and cooled to room temperature. Then, the taste value of cooked rice grain was measured. 

The protein content was measured by using a Kjeldahl nitrogen meter (Foss Tecator, Hoganas, Sweden) according to the operating manual. In terms of total starch content, it was measured with a total starch assay kit (K-TSTA, Megazyme, Wicklow, Ireland) according to the operating manual supported by the kit. 

A rapid viscosity analyzer (RVA; Newport Scientific PTY Ltd., Warriewood, Australia) was used to analyze the viscosity of the rice flour as previously described [19]. Briefly, 3.0 g of rice flour was weighed in an aluminum can with 25.0 g of distilled water. Afterward, the viscosity (centipoise, cP) was determined using a constant paddle at 160 rpm. The protocol was to hold at 50 °C for 1.0 min, increase the temperature to 95 °C in 3.8 min, hold at 95 °C for 12.5 min, decrease the temperature to 50 °C in 3.8 min, and hold at 50 °C for 12.5 min. 

A differential scanning calorimetry (DSC) apparatus (DSC200F3, Netzsch Instruments NA LLC, Burlington, MA, USA) was used to measure starch thermal properties according to Zhang et al. [20]. Briefly, 5 mg starch sample was weighed in an aluminum pan, and 15 μL of deionized water was added. The endothermic curve was determined by heating the pan from 20 to 120 °C at a rate of 10 °C min^−1^. The DSC parameters were obtained from the curve: onset temperature (*T*_o_), peak temperature (*T*_p_), conclusion temperature (*T*_c_), and enthalpy change in gelatinization (Δ*H*).

The digestive characteristics of the rice flour were determined according to the method of Lin et al. [21] with some modification. Specifically, 10 mg of sample was weighed in a 2 mL Eppendorf tube and mixed with 1.7 mL digestion solution (pH 6.0, 200 mM calcium chloride, 0.49 mM magnesium chloride, 8 U pancreatin, and 8 U amyloglucosidase). Afterward, the tube was incubated in a shaking water bath (250 rpm) at 37 °C. At 10, 20, 40, 60, 90, 120, 180, 240, and 300 min, respectively, a 50 μL aliquot of the digestion solution was transferred to a 1.5 mL microcentrifuge tube containing 0.5 mL 50% ethanol to stop digestion. The glucose generated was measured using a glucose oxidase/peroxidase assay kit (Megazyme, K-GLUC) according to the operating manual. The starch digestion curve was calculated following the independent first-order kinetic model. The above experiments were performed in triplicate.

### 2.4. Section Scanning and Transparency Analysis of Milled Rice

Under the environmental conditions of natural light, performance photos of milled rice were taken. Gel Pro Analyzer software was then used to convert the transparency into black-and-white image values; the transparency was positively correlated with the black-and-white image value from Gel Pro Analyzer. The scanning method used for starch grains was similar to that used for rice. However, in the scanning observation experiment of the milled rice section, it was necessary to form a natural section laterally. For the observation of starch granules in the grain cross section, rice grains were broken down naturally, fixed on the sample stage, coated with gold, and observed by scanning electron microscopy (SEM; ESEM XL-30, Philips, The Netherlands). The isolated starch was also analyzed by SEM. The average size and size distribution of the starch grains were determined from the SEM images using Image J software (http://rsbweb.nih.gov/ij/, accessed on 5 May 2020). 

### 2.5. Starch Crystalline Structure Analysis

#### 2.5.1. Powder X-ray Diffraction (XRD) 

A polycrystalline X-ray diffractometer (D8-ADVANCE, AXS Company, Brooke, Germany) was used to evaluate the degree of starch long range order structure according to the method of Wei et al. [22]. Briefly, rice starch samples were pretreated by storing them in a desiccator containing a saturated solution of NaCl for 7 days at room temperature before measurement. The crystallinity is expressed as the percentage crystal area of the total area in the X-diffraction profile. The experiments were performed in duplicate.

#### 2.5.2. Attenuated Total Reflectance-Fourier Transform Infrared (ATR-FTIR)

The external structure of the starch granules was examined via ATR-FTIR spectroscopy on a Varian 7000 FTIR spectrometer with an ATR single-reflectance cell containing a germanium crystal according to the method of Cai et al. [23]. In detail, the starch samples were prepared as described in Section 2.5.1, and the absorbance values for peaks at 1045, 1022, and 995 cm^−1^ were extracted from the ATR-FTIR spectra after correction. The experiments were performed in duplicate. 

#### 2.5.3. Small-Angle X-ray Scattering (SAXS)

SAXS analysis of starch lamellar structures was performed using a Bruker NanoStar SAXS instrument equipped with a Vantec 2000 detector and pin-hole collimation for point focus geometry as previously described by Wei et al. [22]. The starch samples were prepared as described in Section 2.5.1, and the SAXS datasets were analyzed using DIFFRAC^plus^ NanoFit software. The Bragg spacing *d* was calculated from the position of the peak (*q*_o_) according to *d = 2π*/*q_o_*. The experiments were performed in duplicate.

### 2.6. Starch Fine Structure Analysis

The isolated starch was debranched with isoamylase (EC 3.2.1.68, E-ISAMY; Megazyme) before starch fine structure analysis. For the starch relative molecular weight distribution measurement, a PL-GPC 220 high-temperature gel permeation chromatograph system (GPC, Polymer Laboratories Varian, Inc., Amherst, MA, USA) was used according to our previous report [20]. The GPC data used to draw the molecular weight distribution curves was transformed by integral equations based on standard dextran of known molecular weights (2800, 18,500, 111,900, 410,000, 1,050,000, 2,900,000, and 6,300,000). The proportions of amylose (AM), short-chain part of amylopectin (SAP), and long-chain part (LAP) were calculated from the GPC curve. 

The degree of polymerization (DP) of amylopectin was determined using a high-performance anion-exchange chromatography (HPAEC) system (Thermo ICS-5000, Thermo Corp, Sunnyvale, CA, USA) equipped with a pulsed amperometric detector, a guard column, a CarboPacTM PA200 analytical column, and an AS-DV autosampler as described previously [20]. The amylopectin chain distributions were calculated according to the amylopectin cluster model. The above experiments were performed in duplicate.

### 2.7. Data Analysis

The measurement data were collected in Microsoft Excel and analyzed using SPSS 16.0 statistical analysis software. All data were reported as mean ± standard deviation. 

## 3. Results

### 3.1. Analysis of Rice Grain Appearance and Water Content

The grain morphologies of the white rice from the four *indica* rice varieties were compared. As shown in Figure 1A, all rice grains of freshly harvested seeds had transparent endosperms. It should be noted that the glutinous rice YFN showed a good appearance when the water content was high. However, as the water content progressively decreased, the endosperm showed partial opacity (after 4 h of drying) followed by a waxy phenotype (after 6 h of drying). Similarly, with the increase in drying time, the water content of the 28Z variety decreased, and the appearance transparency of grain also gradually showed opacity or translucence. When the moisture content decreased to approximately 11% after 8 h of drying, the translucent phenotype was observed; as the water content decreased further, a stable opaque phenotype was observed. Compared with 28Z, the other two rice varieties, which were HHZ and 9311, showed relatively transparent phenotypes during the dying process. 

To quantify the grain transparency, the endosperm transparency was converted into a relative translucence level by using the gray analysis function in Gel Pro Analyzer. Among the studied varieties, the glutinous rice YFN had the largest relative translucence and highest opacity, while 28Z had a higher relative translucence than HHZ and 9311 (Appendix A). The relationship between water content and relative translucence was evaluated by linear fitting (Figure 1B), revealing a clear negative correlation. The slope of the water-loss curve reflects the difficulty in forming an opaque phenotype; a larger slope indicates greater ease of dehydration and a greater likelihood of an opaque phenotype. The above results indicate that the moisture content plays a key role in the formation of grain transparency, consistent with previous studies [24].

### 3.2. Analysis of Starch Structure

To understand the differences in transparency among varieties, the transverse fracture surfaces of the dry grains and the morphologies of starch particles were observed by SEM. As shown in Figure 2, the glutinous rice YFN showed a typical waxy endosperm, and 28Z exhibited a translucent phenotype under dry conditions. The starch grains of all four varieties were arranged orderly and regularly, although some cavities appeared in the core of a single starch granule of YFN rice (Figure 2(A1,A2)). Several cavities were also observed in a 28Z rice grain when the cross section was frozen in liquid nitrogen; however, the cavity area in 28Z starch grains (2.96 ± 0.01 μm^2^) was significantly lower than that in YFN grains (3.37 ± 0.03 μm^2^; Figure 2(A2,B2)). Few cavities were observed in the cores of the starch granules of HHZ and 9311 rice. Our previous studies have demonstrated a positive correlation between cavity size and grain transparency [20,24]. To further understand this phenomenon, the transparency calculated from the grayscale value of the SEM image was used for correlation analysis. As shown in Figure 1C, the rice grain relative transparency showed a strong positive correlation with the cavity size within the starch granules, consistent with our previous findings. The morphologies of isolated starch grains were analyzed by SEM, and all starch samples showed regular shapes with smooth surfaces (Figure 2(A3–D3)). However, significant differences in the starch granule size distribution were found among the four samples. Based on the starch particle size distributions, the average particle sizes differed among the varieties (Figure 2(A4–D4)). All samples showed unimodal size distributions. The particle sizes with the greatest proportions were 5.4, 4.15, 4.7, and 4.85 μm for 28Z, 9311, YFN, and HHZ rice, respectively (Figure 2(A4–D4)). 

### 3.3. Eating and Physicochemical Characteristics of Rice

In addition to grain moisture content and starch granule structure, other factors such as AC have important effects on rice grain transparency. Thus, the physical and chemical characteristics of the rice grains were further investigated. As shown in Table 1, among the rice varieties, YFN showed the lowest AAC followed by 28Z, 9311, and HHZ. Thus, the low AAC of 28Z might be the primary cause of the translucent grains. The taste value of cooked rice was then measured. Grains from 28Z had a significantly higher taste value than HHZ and 9311 grains, consistent with the fact that lower AC usually corresponds to good taste [4]. No significant differences in moisture content, crude protein content, and total starch content were found among the four samples (Table 1). During the in vitro digestion experiment, the LOST curve of rice starch showed a linear relationship with the digestion rate constant K, indicating that the digestion of rice flour in vitro was a single-phase process. We also investigated the starch function profiles by measuring the starch digestion curves in vitro. As shown in Figure 3, the digestion rate of YFN tended to be stable during the digestion process, whereas the digestion of the other changed to a certain amplitude. The differences in digestion among the three non-waxy varieties became obvious after 90 min. During the first 10–90 min of digestion, the digestion rate (K value) of 28Z starch was not significantly different than that of HHZ and 9311. However, considering the entire digestion process, the digestion of 28Z was relatively slow compared to the digestion of HHZ and 9311. 

### 3.4. Analysis of Starch Pasting and Thermal Properties

Rapid viscosity analysis is often used to analyze the viscosity of starch and can also indirectly reflect the eating quality. The pasting characters of flours from the four rice varieties were measured using an RVA. As shown in Figure 4A, the normal glutinous rice (YFN) had the lowest viscosity among the varieties. The RVA curves of the three non-waxy varieties were similar, although the curve of 28Z showed a lower-viscosity tail at the end of the analysis. The RVA parameters are summarized in Appendix A. The setback visibility (SBV), breakdown viscosity (BDV), and cool viscosity of 28Z were significantly different from those of the other three varieties. The relatively high BDV and low SBV of 28Z are indicative of good eating quality [4,11]. These RVA parameters of 28Z are consistent with previous studies that found that rice with low AC usually have high BDV and low SBV [16,20]. 

To further examine the differences in starch physicochemical properties among the varieties, the starch thermal properties were analyzed by differential scanning calorimetry (DSC) (Figure 4B). The DSC curves of the four varieties were similar, although the curve of YFN showed a delayed gelatinization peak. However, significant differences in the DSC parameters were found between 28Z and the other three varieties. Compared with 9311, the absorption peak of 28Z was delayed, and the initial temperature, peak temperature, and final gelatinization temperature were higher. During gelatinization, the temperature span of 28Z became wider, indicating that this variety requires more energy to complete gelatinization, and that the gelatinization time is slightly longer compared to the other varieties (Appendix A). 

### 3.5. Microstructures of Starch Granules 

The physicochemical properties of starch are often closely related to the starch crystal structure. Thus, the microstructures of starch grains were analyzed by X-ray diffraction (XRD). As shown in Figure 5A, all samples exhibited similar diffraction peaks with strong signals at approximately 15° and 23° and unresolved doublets at approximately 17° and 18°, indicative of the typical A-type XRD curve of most normal cereal starches [25]. Obvious differences were found in the relative crystallinities of the four varieties (Appendix A). Among the varieties, YFN exhibited the highest relatively crystallinity (34.08%) followed by 28Z (27.02%). The structures of the starch lamellae were analyzed by SAXS to better understand the characteristics of the semi-crystalline starch growth rings with thicknesses of 9–10 nm. As shown in Figure 5B, no significant differences between varieties were observed between the half peak width (Δ*S*), the peak position (*S_max_*), and the Bragg spacing (*D*) of starch. Meanwhile, the diffraction peak intensity (*I_max_*) of YFN (336.51 counts) was higher than that of the other three *indica* rice varieties. The *I_max_* value of 28Z was smaller than that of HHZ (*I_max_* 165.85 counts) but higher than that of 9311 (*I_max_* 149.606 counts). The short-range ordered structures of starch granules were analyzed by ATR-FTIR. The degree of short-range order was analyzed by calculating the ratio of starch absorption at 1045 cm^−1^ to that at 1022 cm^−1^ ((1045/1022) cm^−1^). The infrared absorption curves of the four *indica* rice varieties were relatively consistent. Based on the calculated eigenvalues, there was no significant difference in (1045/1022) cm^−1^ among the three non-waxy varieties (Appendix A). However, there was a significant difference in (1022/995) cm^−1^ and the amorphous part of 28Z was less than that of HHZ. 

### 3.6. Starch Fine Structure 

To better understand the differences in grain quality and starch physicochemical properties, the relative molecular weight distributions were measured by GPC. According to the relative area of each GPC peak, the fine-structure characteristics of starch were studied quantitatively. As shown in Figure 6A, three distinct GPC peaks were observed, including two branching peaks, namely, the short-chain peak (Peak 1) of amylopectin, the long-chain peak (Peak 2) of amylopectin, and the amylose peak (Peak 3, or AM). There was no significant difference in Peak 2 of the first-order branching peak (Appendix A). The B-chain content of secondary branch starch was 62.56% for 28Z, significantly less than that of 9311 but more than that of HHZ, although the difference was not significant. After the rice starch was hydrolyzed by amylase, the distribution of amylose chain length was mainly reflected by the spectral peak distinguishing feature of Peak3, which consisted of long amylose (AM1) and short amylose (AM2). The content of HHZ was significantly higher than that of 28Z and 9311, which was mainly reflected in the content of AM1. There was little difference in the content of AM2 between varieties, although the long-chain amylose content of 28Z was greater than that of HHZ and 9311. The branching degree of amylopectin was measured by the area ratio of the short-chain part of amylopectin (SAP) to the long-chain part (LAP). The higher the proportion (SAP/LAP), the higher the branching degree of starch. However, the degree of starch branching (SAP/LAP) differed among the four *indica* rice varieties. The branching degree of 28Z is different from that of HHZ (SAP/LAP) 2.89, but there is a significant difference, YFN and 9311 (SAP/LAP) compared. To further clarify the distributions of amylopectin chain length in the four *indica* rice varieties, the structural features of amylopectin were analyzed by high-performance anion-exchange chromatography (HPAEC; Figure 6B–D). The degree of amylopectin polymerization was higher in 28Z than in 9311 and HHZ, composed of more medium and short chains ΣDP (16–23), reduced ΣDP (25–34), and ΣDP (42–50) (Appendix A).

## 4. Discussion

Rice transparency is an important property in high-quality rice, and its formation mechanism has attracted considerable attention. Rice grain transparency can generally be divided into seven levels under normal dry conditions (in order of decreasing transparency): fully transparent glassy, transparent, translucent, dark, partial white, light milky white, and pure milky white [26]. Rice AC is the key determinant of transparency, and grain transparency decreases with AC. By using rice varieties with different ACs, Zhang et al. found that the grain transparency was positively correlated with the grain moisture content, while it was significantly negatively correlated with AC and starch cavity size [20,24]. In the present study, we also found a strong linear relationship between rice transparency and water content (Figure 1B). Thus, controlling the moisture content is a promising approach to maintaining grain transparency. For example, soft rice with low AC is usually packaged under vacuum to maintain a transparent endosperm with certain moisture content. However, under normal conditions, the loss of water occurs continuously, and grains with relatively low AC will exhibit a dull grain phenotype. We found that the AC threshold below which rice grains remain transparent is approximately 14.0%; thus, rice grains with AC below 14% will gradually lose transparency as water is lost [11]. In the present study, 28Z rice had a relatively low AC of 13.05%, close to the threshold value of 14%. The 28Z rice grains were translucent with a better appearance than soft rice varieties with ACs below 12% and good eating quality. This moderate AC of 28Z represents a good trade-off between rice grain appearance and eating quality.

In addition to grain moisture content and AC, starch morphology is an important factor in endosperm transparency. Liu et al. found that the small cavities between starch granules and the holes within starch granules contribute to the opaque endosperm of soft rice [14]. Some low-transparency rice endosperms with chalky or silty (or floury) phenotypes were found to have loosely arranged starch grains, small starch particle volume, and spherical or irregular grain structures [16,27]. However, in the present study, the cross-sectional SEM images of the four *indica* rice varieties revealed closely arranged and regular starch grains with no holes between starch granules but remarkable cavities within single starch granules. These cavities appeared even in the transparent grains of HHZ and 9311 under dry conditions. The cumulative cavity area of 28Z was lower than that of YFN but greater than that of HHZ and 9311. Through correlation analysis, we found that grain transparency was significantly negatively correlated with the cavity area within starch granules. These results are consistent with our previous findings that the cavity size in the starch granule core adversely affected the opaque appearance [20,24]. Studies on other plant starches such as canna and wheat also revealed cavities within single starch granules [28,29]. Considering that amylose is mainly concentrated in the centers of granules, amylose may form a stable and well-distributed structure with both water and amylopectin during the arrangement of the starch semi-crystalline structure [29,30]. Thus, the cavities within starch granules result from the retraction of the semi-crystalline structure during water loss, resulting in the translucent or dull appearance of the rice endosperm. 

The AC within the rice endosperm is the most critical determinant of rice eating quality along with the physicochemical properties of starch [31]. Generally, AC is negatively correlated with taste score and positively correlated with the hardness of cooked rice [31]. In the present study, the AC of 28Z was lower than that of HHZ and 9311, indicating the better taste of 28Z. Rice varieties with low AC often have large BDV, small SBV, and low recovery value [32,33]. Sasaki et al. [34] showed that decreasing the AC increases peak visibility and decreases final visibility and SBV, potentially because low AC inhibits starch expansion and the rate and degree of starch recovery [35]. Thus, the relatively high BDV and low SBV of 28Z rice suggest good taste quality. In fact, the crude protein content and GT are also important factors affecting rice eating quality; however, in the present study, there was no significant correlation between these two parameters in the three non-waxy varieties. Therefore, the low AC of 28Z rice is the main reason for its good taste value. 

The physicochemical properties of starch are often closely related to the fine structure of starch. In the four rice varieties, the relative crystallinity of starch decreased with the AC values. In the XRD pattern, the peak at the 2θ angle of 23° corresponds to the amylose–lipid complex [36]. The diffraction peaks of YFN and HHZ appeared earlier than those of 28Z and 9311, indicating that the contents of amylose and lipid increased gradually from the inner to outer regions of the grains. The contents of amylose and lipid in 28Z were lower than those in HHZ and 9311, although the differences were not significant. Thus, the decrease in starch crystallinity was caused by the increase in amylose [30]. SAXS analysis revealed higher-intensity diffraction in YFN compared to the other three *indica* rice varieties, while the diffraction peak intensity of 28Z was smaller than that of HHZ and 9311. Yuryev et al. reported that *I_max_* was negatively correlated with AC [20]. Therefore, we can deduce that the AC of 28Z is between that of HHZ and 9311; thus, compared with HHZ, the appearance of 28Z is less transparent, and the lamellar structure of starch crystal particles forms less easily than in HHZ and 9311. Sevenou et al. found that (1022/995) cm^−1^ can be used to reflect the ratio of amorphous to ordered components in starch [37]. In the present study, the rice varieties exhibited significantly different (1022/995) cm^−1^ values, and 28Z had a smaller amorphous component than HHZ. This is consistent with the crystal structure analysis of starch by XRD and SAXS.

Generally, rice starch consists of two components, amylose and amylopectin, and the fine structures of both components contribute to physicochemical properties. The relative molecular weight distributions determined by GPC showed that the B-chain content of secondary branch starch in 28Z was significantly lower than that of 9311 but higher than that of HHZ. Rice varieties with better eating quality generally have a higher proportion of short-chain amylopectin (AP1) [16,38]. The DP of the amylopectin structure affects the starch crystallinity and gelation properties [4]. Jane et al. reported that the crystalline lamellae of starch are mainly composed of medium and long chains, and the presence of shorter-chain DP6-11 will reduce the crystallinity of starch [39]. Therefore, the crystallinity of 28Z starch was higher than that of HHZ and 9311.

In summary, we compared the appearance, physicochemical properties, and starch fine structures of rice grains from four *indica* rice varieties. Notably, 28Z is a novel type of low AC rice with a higher AC than normal soft rice varieties, resulting in a translucent endosperm rather than a dull appearance. Our results indicate that grain transparency is positively correlated with water content and negatively correlated with the cavity size within starch granules. In addition, among the four *indica* rice varieties, we found that the eating quality of 28Z rice was better than that of non-glutinous rice. Considering the moderate AC and good taste value of 28Z, we believe that 28Z is a promising germplasm to improve rice grain quality through breeding. In particular, this variety achieves a trade-off between rice eating quality and appearance. 

## Figures and Tables

**Figure 1 foods-11-01378-f001:**
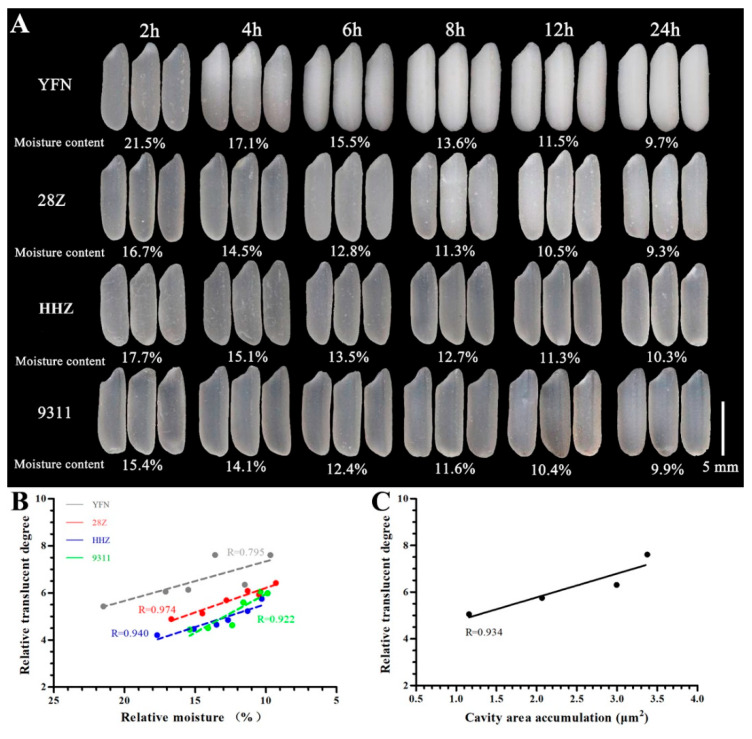
Appearance of rice grains with gradient drying and correlation analysis between grain translucency and moisture and between grain translucency and cavity area within starch granules. (**A**) Grain morphology after drying for 2, 4, 6, 8, 12, and 24 h at 40 °C. (**B**) Correlation between grain translucency and moisture content of milled rice. (**C**) Correlation between grain translucency and cavity area within starch granules.

**Figure 2 foods-11-01378-f002:**
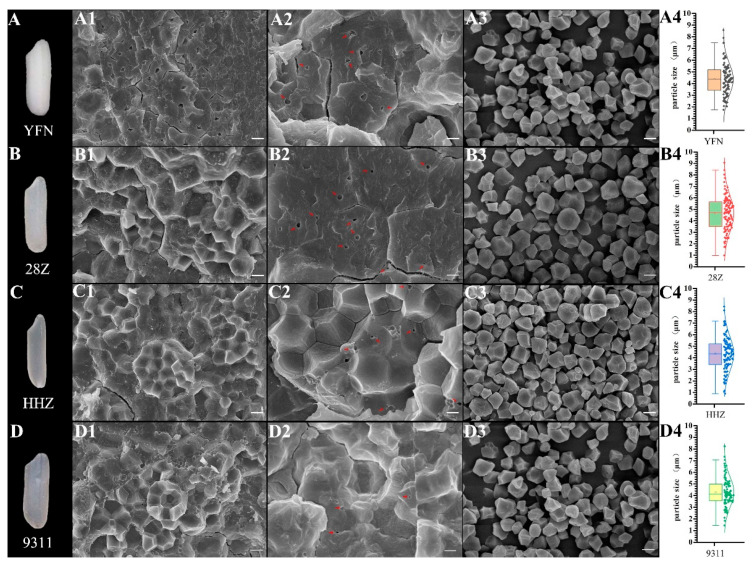
Scanning electron microscopy (SEM) images of grain cross sections and starch granules. (**A**–**D**) Grain morphologies at a moisture content of approximately 10%. (**A1**–**D1**,**A2**–**D2**) SEM micrographs of grain cross sections (scale bar = 10 μm). Arrows indicate air spaces within starch granules. (**A3**–**D3**) SEM micrographs of isolated starch granules (scale bar = 5 μm). (**A4**–**D4**) Starch granule size distributions (*n* = 200).

**Figure 3 foods-11-01378-f003:**
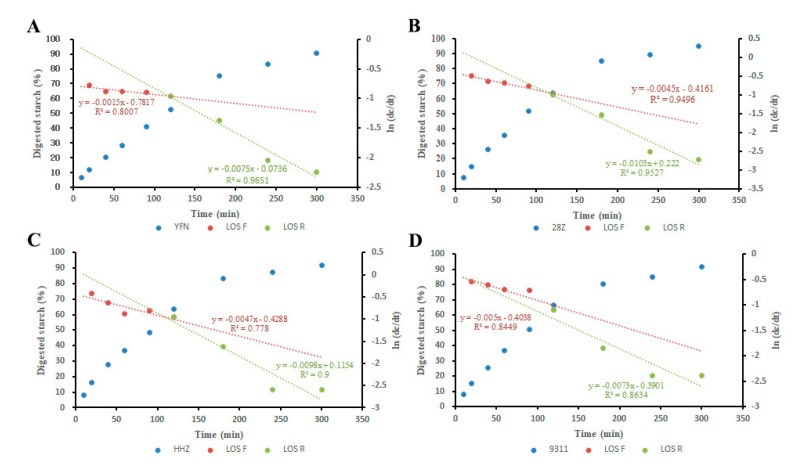
In *vitro* digestion characteristics of rice flour. Starch digestion curves and LOST plots of endosperm starch in YFN (**A**), 28Z (**B**), HHZ (**C**), and 9311(**D**) seeds. Los F and Los R represent the digestive LOST curve from 10–120 and 120–300 min, respectively.

**Figure 4 foods-11-01378-f004:**
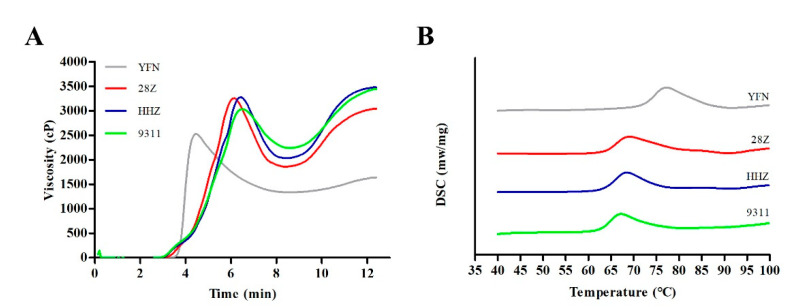
Pasting (**A**) and thermal (**B**) properties of rice flours from different rice varieties.

**Figure 5 foods-11-01378-f005:**
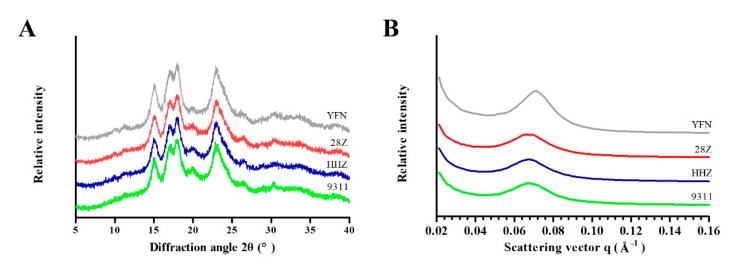
X-ray diffraction (XRD) patterns (**A**) and small-angle X-ray scattering (SAXS) profiles (**B**) of rice starches from different varieties.

**Figure 6 foods-11-01378-f006:**
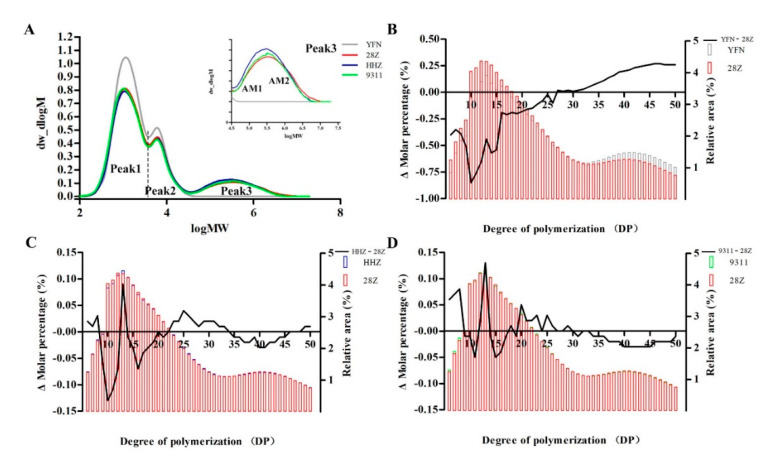
Fine structure of de-branched starches from four rice varieties determined by gel permeation chromatography (GPC) and high-performance anion-exchange chromatography (HPAEC). (**A**) Relative molecular weight distributions of de-branched starch. (**B**–**D**) Chain length distributions of amylopectin. MW, weight average molecular weight; AM, amylose; DP, degree of polymerization.

**Table 1 foods-11-01378-t001:** Statistical analysis of the taste and physicochemical properties of four *indica* rice varieties.

Rice Variety	AAC (%)	TV (Points)	MC (%)	PC (%)	TS (%)
YFN	4.64 ± 0.74 d	—	11.87 ± 1.58 a	9.38 ± 1.22 a	76.8 ± 2.1 a
28Z	13.05 ± 0.14 c	83.34 ± 1.81 b	10.94 ± 0.41 a	10.44 ± 1.21 a	76.1 ± 2.1 a
HHZ	16.16 ± 0.22 a	78.05 ± 3.41 a	12.03 ± 0.70 a	9.50 ± 1.07 a	77.9 ± 1.7 a
9311	15.11 ± 0.18 b	76.54 ± 3.24 a	11.04 ± 0.55 a	9.87 ± 0.76 a	77.7 ± 2.2 a

Values are means ± standard deviation (*n* = 3). Values in the same column with different letters are significantly different based on Student’s *t*-test (*p* < 0.05). AAC: apparent amylose content; TV: taste value; MC: moisture content; PC: crude protein content; TS: total starch.

## Data Availability

The data presented in this study are available on request from the corresponding author.

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
