# Peer review of "The Underlying Physicochemical Properties and Starch Structures of indica Rice Grains with Translucent Endosperms under Low-Moisture Conditions"

_foods, 2022, doi:10.3390/foods11101378_

Round 1

Reviewer 1 Report

In this paper  the authors explore the grain transparency, psyhochemical characteristics, eating quality, morphology and fine structure of starch of four selected  indica rice varietes, under low moisture conditions. The work is well elaborated, the analyzes are detailed and adequate and the results are presented at a satisfactory level.

The comments are as follows:

The authors should better indicate the purpose of the work, because it is not clear from the  introduction and discussion. Is the purpose an investigation of indica varieties transparency, or promoting the 28Zhan landrace? Are there any papers that have been studied indica varietes? It is only mentioned japonica rice. If there, mention this papers ang linkage the obtained results with literature data.

Line 44, write japonica in italic letters

Line 78, In which period was the maturity and harvest if they were planted in summer?

Explain the difference between glutinous (also no gluten) rice variety and non-glutinous varieties in nutritive, agronomical and morphological sense.

Line 85, Type of oven?

Table 1. Why YFN was not analyzed for taste?

Check the numbering of subsection. 3.2 is repeated, 3.7 is wrong numbered

Author Response

Dear review experts,

 I sincerely thank review experts  for their wonderful comments and put forward very constructive suggestions and opinions to us. We carefully reviewed and revised the manuscript. We believe that under the guidance of our beloved editors and peer reviewers, through our careful modification, we can make a good answer to the review experts' comments. Please refer to the attachment, and you can see our answer from the attachment.

Through our careful revision and improvement, we hope that the revised version can meet your requirements and enable the manuscript to be published in the form of food. Thank you again for your wonderful comments and for spending a lot of valuable time on our manuscript. Finally, give my most sincere wishes for you.

Sincerely yours,

Fei Chen

College of Agriculture, Yangzhou University

Yangzhou 225009, Jiangsu, China

Reviewer 2 Report

key words - I suggest "starch structure" as important, because I should rank second in the title of the work. Consider removing the word "taste"

line 31- I think that in the introduction you should not introduce AC abbreviations immediately and further develop in this part. it should be easily readable text to the recipient without having to memorize abbreviations. Of course, in the part of the methodology - discussion and discussion of the results - it is most justified

line 45 "we generated" - in this chapter we refer to researchers' results, an overview and the current state of knowledge. I believe that the authors should speak of the results of their research in the third person

the aim of the study is not clearly stated, the authors mention the results of earlier research and over-generalize the purpose of this research

2.3. Physical and chemical quality analyzes - the authors exaggerated too much with the combination of such different parameters indicating the quality of rice. Each of the analyzed physical or chemical quantities should be described in a separate section. The authors shortened the methodology too short, it should be described in detail so that the potential reader could repeat it under their own circumstances and consciously discuss it with these and theirs. Such "hiding" the methodology may, in extreme cases, result in analytical or interpretative errors. Besides, not all readers have access (or when using the results of the work) know that there is a barrage of methodologies. A must to refill !!!!!

2.5. Starch crystalline structure analysis - should contain 3 sections with a full description of the measurement performance 2.5.1. DSC ...... 2.5.2 ATR-FTIR .. 2,5,3, SAXS...

2.6. detail the methodology !!!!!!!!!!!!!!!!!!!!!!!!

2.2. Rice flour and starch preparation and moisture content determination - I seriously wonder if the mention of "moisture content" here is correct. From the results it can be seen that the authors discuss the moisture level as a function of time and temperature. drying. Maybe the "sorption properties of native starch" would be better ... because it's wrongly titled

A clearly separated chapter "conclusions" !!!
  The authors have access to really good measuring devices and they present the results of the work very nicely, but for incomprehensible reasons (I can guess what) they hide the assumptions and parameters of the work - after completing it, it will be a valuable work.

Author Response

(The authors gave the same response as above.)

Round 2

Reviewer 2 Report

In view of the revised version of the article submitted, I see a clear improvement in its quality mainly related to the refinement of the methodology and the corrections indicated in the introduction. I think it would be a big plus to include applications, but I know that according to the publisher, they are optional. The work is suitable for publication in this form.